# Impact of health systems interventions in primary health settings on type 2 diabetes care and health outcomes among adults in West Africa: A systematic review

Eugene Paa Kofi Bondzie[1]*, Kezia Amarteyfio[1], Yasmin Jahan[2], Nana Efua Enyimayew Afun[1], Mary Pomaa Agyekum[3], Ludovic Tapsoba[4], Dina Balabanova[2], Tolib Mirzoev[2], Irene Agyepong[1]

**1** Ghana College of Physicians and Surgeons, Accra, Ghana, **2** London School of Hygiene and Tropical Medicine, London, United Kingdom, **3** Dodowa Health Research Centre, Accra, Ghana, **4** Catholic University of West Africa, Bobo-Dioulasso, Burkina Faso

☯ These authors contributed equally to this work.

* eugenebondzie@gmail.com

## Abstract

Type 2 diabetes is ambulatory care sensitive and adequate outpatient primary care supported by strong functional health systems can reduce avoidable complications and related mortality. A large body of published evidence exists on pharmacological agents and non- pharmacological interventions for the management of type 2 diabetes. However, the evidence on health systems' ability to support patients' primary care needs, especially in West Africa, where non-communicable disease (NCD) is an increasingly important part of the disease burden is uncertain. This systematic review explores the current published evidence on health systems interventions to support primary health facilities for type 2 diabetes care and impact on health outcomes, service access and quality in West Africa. The World Health Organization health systems building blocks and other post building blocks health systems frameworks guided our search and analysis. Only three pilot studies, including two randomized controlled trials and one pre-post study, met all our first inclusion criteria. However, we included 12 other studies which did not meet all the inclusion criteria but reported on a health system intervention for complete analysis: (The criteria were expanded to include studies conducted outside primary care settings. The rationale was that findings from such studies may influence primary care. Also, non-randomized control trials were later included). Our results showed that interventions with significant impact on glycemic control, treatment adherence, health literacy, and other associated outcomes addressed intersections between the individual health system blocks/areas. Thus, four cross-cutting themes related to the building blocks were found during analysis. The first theme was on interventions targeting the availability of trained health workers and the quality of their services; the second was on interventions targeting institutional infrastructure and resources for management; the third was on interventions targeting leadership and organizational culture and; the fourth was interventions targeting relationships

**Data availability statement:** We contacted the corresponding authors of relevant studies for which data related to study methods and outcomes were unclear or missing via emails, providing a 4-week timeline for response. In cases where authors did not respond, we proceeded with data synthesis and clearly reported the missing information and its potential impact on the overall findings in the limitations section. The absence of this information does not compromise the core dataset necessary to reproduce our findings. All relevant data, including the minimal dataset, are available within the paper and its Supporting Information files.

**Funding:** This research was conducted as part of the health research study on strengthening capacity for Non-communicable disease (NCD) control in West Africa (Stop NCDs project) funded by the National institute for health Research (NHIR), global health research centres: research and institutional capacity strengthening In NCDs Call 1 under Grant number- 203246 (http://www.nihr.ac.uk/ccf), awarded to IA and TM. The funders had no role in study design, data collection and analysis, decision to publish, or preparation of the manuscript.

**Competing interests:** The authors have declared that no competing interests exist.

among stakeholders. A fifth theme highlighting patients and family empowerment for type 2 diabetes control was also found in most of the interventions.

## Conclusion

These findings confirm that the rising burden of non-communicable diseases in West Africa, particularly type 2 diabetes, can be tackled by strong functional primary healthcare systems.

## Introduction

Type 2 diabetes, formerly known as adult-onset diabetes, is a major global public health problem [1]. It is characterized by high blood glucose, insulin resistance and a relative lack of insulin and it manifests with symptoms of increased thirst, frequent urination, weight loss and sometimes increased hunger [2]. Long-term complications include ischemic heart disease, retinopathy, nephropathy and limb amputations [3]. Until recently, it was thought to affect adults who were middle-aged, or older but contemporary trends have shown an increased incidence in young people [4]. Reports from epidemiological studies suggest that adults aged 20–79 years living with type 2 diabetes are about 10.5% of the world's population [5]. In Sub-Saharan Africa, the disease is estimated to increase by 129% by 2045 due to an increase in population and rapid urbanization [5]. This phenomenon is worsened by the substantial number of undiagnosed individuals living with the disease. According to the International Diabetes Federation (IDF) Atlas, people with undiagnosed diabetes in the African region represent the highest comparative proportions worldwide, currently at 54% and expected to increase by 2045 [5]. Consequently, the number of diabetes-related deaths was reported at 416,000 compared with 111,100 deaths in Europe. West African countries have estimated prevalence rates of 3.7% in Nigeria [6], 3.95% in Ghana [7] and 1.7% in Burkina Faso [8]. Despite these numbers, glycemic control is still sub-optimal [9–11]. Regulating blood glucose concentration is essential to prevent severe complications [12]. Type 2 diabetes mellitus is ambulatory care sensitive and good primary care can prevent these complications, avoid premature death, and ensure a reasonable quality of life [13,14]. Studies suggest that offering prompt referral and reviews to patients with ambulatory care sensitive conditions (ACSC) decreases the rate of complications and emergency visits [15,16].

Though a large body of published evidence exists on pharmacological agents and non-pharmacological interventions in regulating blood glucose concentration, the health systems' ability to support high-quality primary care in meeting patients' needs may be inadequate [2,17–21]. The essential management of non-communicable diseases (NCDs), particularly diabetes, has been postulated to depend largely on functioning health systems [22]. Effective primary healthcare systems can improve the delivery of diabetes care and promote patient's access and use of quality services including access to medications, health facilities and specialists, with an overall impact on glycemic control and other associated outcomes [20,23,24]. The primary healthcare system in West Africa is oriented towards managing communicable diseases, while NCD management seems less prioritized [9–11]. Healthcare policies and intervention programs mainly exist on paper in many parts of the region [25–27]. Primary health care facilities are ill-prepared to implement essential interventions for NCD control. Results from assessment surveys show a lack of essential medicines, basic equipment and diagnostics; lack of qualified trained personnel; and poor referral systems for the management of diabetes [28–31].

The health system according to the World Health Organization (WHO) consists of all organizations, people and actions whose primary intent is to promote, restore or support health [32]. There are several frameworks proposed for describing and analysing health systems in the literature. Across these frameworks, "health" of individuals through life, often measured in terms of fatal and non-fatal outcomes, is considered as the defining goal of health systems. The WHO building blocks framework (Fig 1), identified as key in determining the ability of health systems to deliver its intrinsic goal of health, uses resources such as human, infrastructure, equipment, tools and supplies; and financing arrangements, health information systems, governance or stewardship, medicines and technology to summarize the health system. [33] Subsequent frameworks [34–38] have built on and carried the conceptual thinking further. The framework by Witter et al, describes mechanisms of change within the health system blocks/areas, suggests the implementation process goals and outlines the final desired outcomes (Fig 2) [38]. Also, increasingly recognized as key in the determinants of the functioning of health systems are people within health systems their power and how they choose to exercise it; and the processes they set up and use to run the health system [39]. People are central to and drive health systems like the way in which the software drives computer systems or hardware. Power and influence in organizations are "the capacity to effect (or affect) organizational outcomes"; "to get desired things done, to effect outcomes – actions and the decisions that precede them"[39] In health systems, depending on people, their power, interests, networks, relationships etc., there will be variation in performance even with the same hardware or building blocks. Thus, the strength of the health system can be attributed to synergistic effect of the building blocks which serve as the hardware components of the computer, and the people, processes and power which serve as the operating systems and software on which the computer runs smoothly.

In this systematic review, we drew on the WHO six building blocks as an analytical framework to describe the health system, in six core functional components/areas of service delivery, health workforce, supply chain, health information systems, financing and governance [33]. Evidence suggests these core components are useful for assessing the process of strengthening the health systems [40,41]. Strategies that promote any of these components of the health system or across integration of the domains are described as health system intervention and have been proven to have significant impact on the outcome of a health condition in the African setting [40]. In addition, we drew on post building block frameworks, applying the

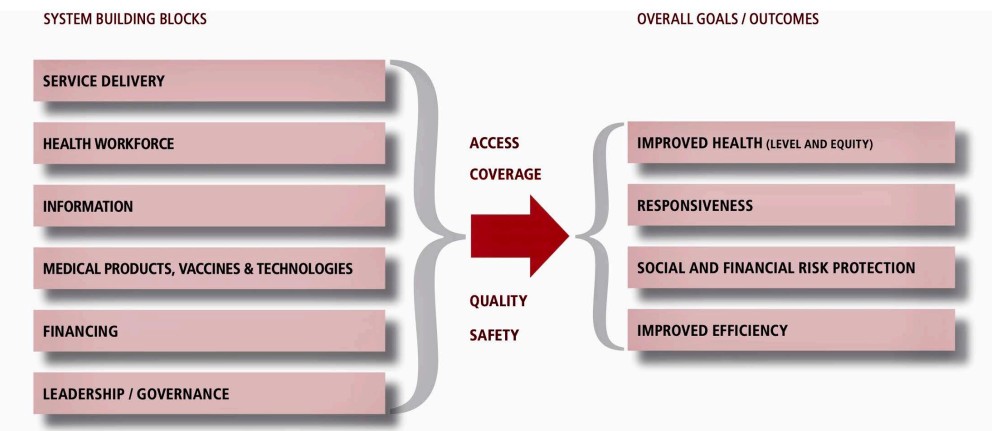

**Fig 1. Witter et al, 2019 health systems framework [23].**

mechanisms of change in the Witter et al framework and a descriptive approach by Byiringiro et al [40] to define the health system interventions and health outcomes.

This current systematic review provides a summary of established interventions/ pilot interventions to enhance the availability, delivery, accessibility and quality of primary health-care for type 2 diabetes in West Africa; and explores the effectiveness of these system-level interventions on some patient outcomes; glycemic control, disease awareness and treatment adherence.

Results from this study feeds into strengthening or co-production of interventions for NCDs in three West African countries; Ghana, Burkina Faso and Niger, as part of the ongoing Project for NCD Control in West Africa (Stop NCD Research).

## Objectives

1. To identify the health system interventions that influence the availability, accessibility, delivery and quality of type 2 diabetes care among adults in West Africa.

2. To explore the evidence on the effectiveness of these system-level interventions on glycemic control, awareness and treatment adherence.

3. To explore the impact of these system-level interventions on any associated health outcomes (e.g., reduction in diabetes complications, financial risk protection etc.)

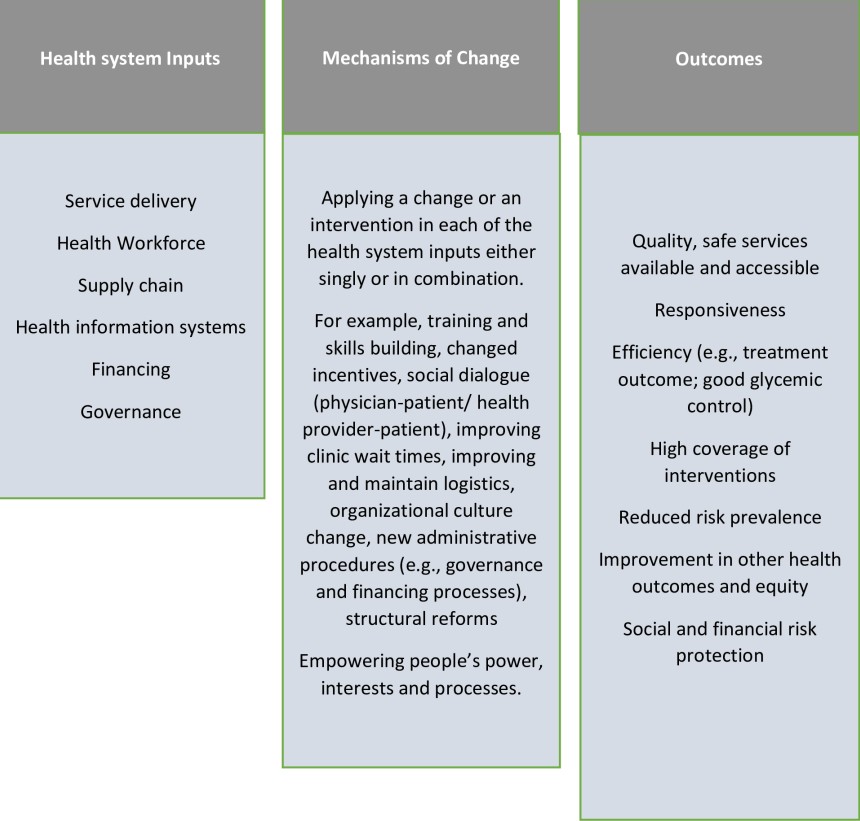

**Fig 2. The WHO health systems framework. Image credit: WHO website [38].**

## Methods

We conducted this review following the reporting guidance provided in the Preferred Reporting Items for Systematic Reviews and Meta-Analyses (PRISMA) (S1 File) [41] between August 2023 to March 2024. The study protocol [42]was registered on PROSPERO (Reg. number: CRD42023439897) (S1 Text).

### Inclusion and exclusion criteria

Studies were eligible for inclusion if they;

- Were conducted among West African adults (aged 18 years or more) with type 2 diabetes,

- Reported on already set up interventions/ pilot interventions targeted at improving the availability, delivery, accessibility or quality of health care for type 2 diabetes,

- Explored the impact of interventions on glycemic control, disease awareness, treatment adherence or associated healthcare outcomes.

- We first included only articles with robust study designs (randomized control study designs (RCTs) clinical control trials (CCTs) or pre-post/quasi experimental study designs) to show causality. Other studies (Cross-sectional studies, cohort studies, case-control studies etc.) that evaluated a health system intervention were later included in our expanded scope and the explanation provided for such inclusion.

- Articles published in English or French from the year 2000 were eligible. This publication time limit was selected to encompass studies that would have likely benefited from recent developments and improvements in RCTs, CCTs and quasi-experimental studies.

- Case series and gray literature (e.g., books, commentary, dissertations, conference proceedings, modeling and simulation studies) were excluded.

- Studies on gestational diabetes (since its natural history varies significantly from type 2 diabetes), and type 1 diabetes were excluded.

### Description of interventions

The interventions were considered as system-level if they addressed any, or all the health system areas described by the health system building blocks in the following ways;

- Service delivery. We considered the studies to address this aspect of the health systems if the target of the intervention was to enhance patients access to health services for diabetes like screening, health education, treatment, and follow-up, either through equitable distribution of care services, reduction of out-patient waiting time, revision of time allocated for services, or integrating the delivery of diabetes care with other established health services like HIV or TB.

- Health workforce/human resource. Interventions of interest include strategies to increase the number of providers, improve provider knowledge and implementation of diabetes management guidelines, address provider decision support systems, promote teamwork, and institute task sharing and/or task-shifting strategies to include providers who do not normally perform certain diabetes management tasks like prescribing medications.

- Supply chain. In this aspect of the health systems, we looked to include studies with interventions that target enhancing the ability of the procurement systems to ensure the availability of anti-diabetic medications, availability and maintenance of calibrated glucometers,

availability of consumables to conduct screening and other diabetes investigations, patient follow-up technologies like use of short message systems, and treatment guidelines.

- Health information systems. Interventions that explore the use of patient registries and information between patients and providers and among providers were considered.

- Financing. Interventions aimed at reducing patients' out-of-pocket spending or funding of diabetes care at the health facilities through national and sub-national spending on health insurance premiums, or other relevant financial reliefs were considered.

- Governance. We looked to include studies with interventions to promote a facility's leadership awareness of the burden of poor diabetes management, or if the intervention used strategic planning and implementation of national diabetes management protocols in a health care facility, or explored the accountability measures at the health facility, or applied regular performance.

- Additionally, any interventions that influenced patients'/ peoples' power/interest/ process of decision-making towards health outcomes were considered (e.g., Patient self-management education, relative support etc.). These are interventions that address the centrality of people in health systems.

## Operational definitions

1. Glycemic control. Defined by the American College of Endocrinologists as glycated hemoglobin (HbA1c %) levels below 7% or fasting blood sugar levels below 110mg/dl (6.1 mmol/l) [12].

2. Diabetes awareness. We describe as knowledge about clinically diagnosed diabetes, either with HbA1c or fasting blood sugar levels and knowledge about the natural history of diabetes, including prevention risk factors and complications as well as its management by any person or organization.

3. Treatment Adherence. Defined as consistently following a treatment plan (either an oral anti-diabetic or insulin, or a lifestyle plan) as prescribed/advised/implemented by a health care provider [43].

## Search strategy

Using the following keywords and their medical subject headings ("Health Services," "Health Policy," "National Health Programs," Delivery of Health care," "Primary Health Care," "Health Facilities," "Health Care Facilities, Manpower, and Services," "Healthcare Financing," "Insurance, Health, Reimbursement, "Health Information Systems," "Equipment and Supplies", "Task sharing"); and intersected with type 2 diabetes ("Diabetes mellitus, type 2," "Glycemic Control," "Diabetes Complications," "Hyperglycemia,") and individual West African countries, we searched PubMed, Google scholar and Cumulated Index to Nursing and Allied Health Literature (CINAHL) between January 2000 to January, 2024, and Cairn. info from the start to January 2024 (S2 File).

## Study selection

The literature search results were transported to Rayyan software [44], and the screening of titles and abstracts of articles were conducted by a pair of independent reviewers. Subsequently, conflicts were resolved through discussions. A pair of reviewers conducted full-text screening and final extraction of the included articles.

## Data extraction

A data extraction form was created in Microsoft Excel. A pair of reviewers independently extracted data on the first author, year of publication, country, study design, sample size, sample population, study site, type of intervention, health system domain and intervention explored, and outcomes.

## Dealing with missing data

We contacted the corresponding authors of relevant studies for which data related to study methods and outcomes were unclear or missing via emails, providing a 4-week timeline for response. In cases where authors did not respond, we proceeded with data synthesis and clearly reported the missing information and its potential impact on the overall findings in the limitations section. The absence of this information does not compromise the core dataset necessary to reproduce our findings. All relevant data, including the minimal dataset, are available within the manuscript and its Supporting Information files.

## Risk of bias assessment

The Cochrane risk of bias tool was used to assess the risk of bias and quality of the studies with RCT study design. The studies were scored either as high, low or unclear risk of bias by two independent reviewers based on six domains: selection bias (random sequence generation, allocation concealment); performance bias (blinding of participants and personnel); detection bias (blinding of outcome assessment); attrition (incomplete outcome data); reporting bias (Incomplete reporting); Other bias. The Joanna Briggs Institute quality assessment tool was used to assess the risk of bias in the quasi-experimental studies. Risk of bias in observational studies were assessed using three domains: selection bias; information bias (differential misclassification & non-differential misclassification) and confounding. The risk of bias tools are represented in S3 File.

## Analysis and synthesis of findings

To synthesize our findings, we first examined each intervention against the health system domains/building blocks. The health system building blocks were interdependent; interventions in one block had intended or unintended consequences in another block. The process of examining interventions against the building blocks showed that while the building blocks were useful for initial categorization, they were limited in exploring interventions that were dependent on the actions in multiple health system domains. We then classified interventions that cut across more than one health system block/domain under their overarching themes. Two reviewers discussed and adjusted the final overarching themes based on the numbers of articles identifying a particular theme and the strength of the evidence within those articles. Additionally, a separate theme appeared for interventions that influence people's power and interest in making decisions towards their own health care. The outcomes of the included studies were classified under the following: glycemic control, treatment adherence, disease awareness, and other health outcomes (e.g., social and financial risk protection, quality of life, and protection from complications). We performed an assessment of confidence in the cumulative estimate of effect of interventions on glycemic control, treatment adherence, and disease awareness using the Grading of Recommendations, Assessment, Development, and Evaluation (GRADE) approach [45]. Finally, we explored the link between the interventions and outcomes and compared them where applicable using a narrative synthesis.

## Results

3347 studies were retrieved from the search. After deduplication, we screened 2917 articles based on titles and abstracts. Thirty-four articles were retrieved for full-text reading. After the full-text reading, only 3 studies met the first eligibility criteria [46–48]. However, the scope was widened to include 12 other studies which had some sort of evaluation on a health system intervention, to allow for a complete thematic analysis. These 12 studies first failed the eligibility criteria because some were conducted with less robust study designs and others, though conducted with robust study designs, did not meet the full inclusion criteria because they were conducted in tertiary-level facilities. A total of 15 articles from Ghana, Nigeria, Guinea-Bissau and Mali were thus included in this review (Fig 3). Six of these were RCTS conducted in Ghana (n = (3) [47,49,50], Nigeria (n = 2) [51,52] and Mali (n = 1) [46]. Four quasi-experimental studies were conducted in Nigeria (n = 3) [53–55] and Ghana (n = 1) [48]. Five cross-sectional studies were conducted in Nigeria (n = 4) [56–59] and Guinea-Bissau (n = 1) [60]. The quality assessment showed a low risk of bias in 3 out of the 15 studies. Five studies were graded as "high", and the rest were graded as "unclear". The most common issues on high risk of bias of the RCTS were lack of blinding of participants and personnel, and lack of blinding during outcome assessment. Though it is not workable to blind participants in such health system interventions, a description of the reasons for omitting standard procedures of a study design is necessary. The geographic representation of included studies by country is shown in Fig 4.

### Health systems interventions explored

Following the analytical process described earlier, the review findings were divided into four themes that cut-across the different building blocks of the health system and a fifth theme that underpins patient/client empowerment, a critical intervention that affects how the other systems work. A table of all data extracted from the primary research sources highlighting the findings on interventions for each individual health system building block in the first stage of the analysis and the quality assessment of included studies are presented in S1 Table. The summary of findings of the key cross-cutting themes on health system interventions for type 2 diabetes are presented in Table 1.

Beginning from the first theme, the interventions are presented below:

### Interventions targeting health worker's availability, roles and activities

The first major cross-cutting theme underpins interventions addressing the availability of health workers, their roles and activities towards patient care and improvement of service delivery. This theme highlights interventions that cut-across the health workforce, service delivery supply chain and health information building blocks. Eight studies [48–51,53,54,56,60] explored the impact of the following interventions on our outcome measures; training of service providers on evidence-based practices, provision of educational materials and flip charts for workers, provision of equipment and tools to improve diagnosis and treatment, establishment of health information systems to ease information sharing, adherence to guidelines and provision of tools for follow-up.

Most of these health workers had some background training in primary care but the interventions which focused on setting up multi-professional diabetes teams and task sharing, had significant impact on glycemic and non-glycemic endpoints. In 3 separate pilot studies conducted at health facilities in Nigeria and Ghana, task sharing was recognized as a key strategy to overcome human resource constraint, e.g., nurses prescribing medications in areas where there were no physicians, pharmacists providing diet education to diabetics and nurses providing education for chronic care. Consequently, adequate glycemic control and high

treatment compliance were achieved [49,51,54]. In one study, an electronic health information system was built in a health facility to prompt clinicians about abnormal laboratory results, provide alerts on patients with elevated risk for diabetes complications and notify physicians to adhere to treatment guidelines. This intervention led to significant improvement in fasting plasma glucose and other cardiovascular risk factors [50].

## Interventions targeting institutional infrastructure and resources for management

The second key theme that emerged underpins interventions addressing physical structures, commodities and funding. These interventions particularly address not only the service delivery health system block but also finance, leadership and supply chain.

Space allocation to set up diabetes clinics within existing health facilities and the provision of diabetes test kits were some of the significant interventions found within the studies under this theme [48,60]. Allocating specific clinic days and hours for diabetic patients led

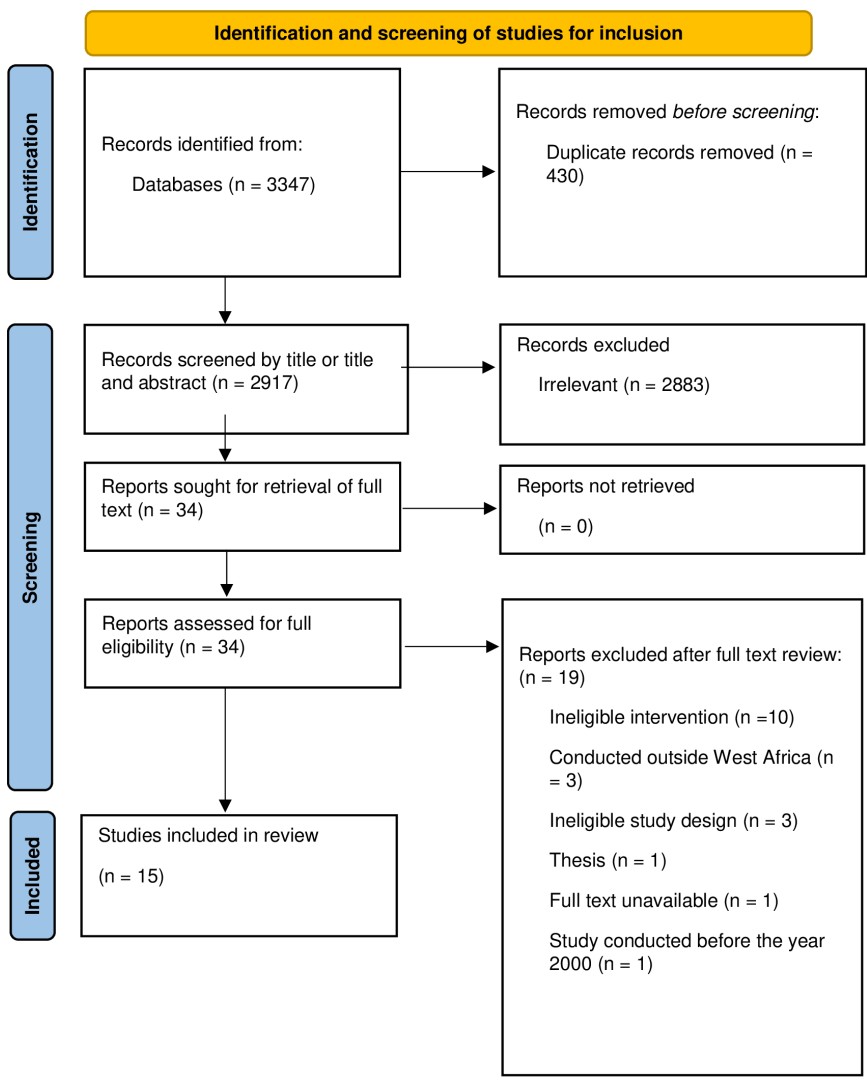

**Fig 3. PRISMA table of all studies identified in the literature search including those that were excluded from the analysis.**

to an increase in patient-physician contact time, reduction of patient waits times and overall, improvement in compliance to clinic appointments.

Health insurance was another important intervention. Patients who were on health insurance and subsidized care plans had better adherence to medications and glycemic control compared to patients who paid out-of-pocket. However, health insurance was only limited to a few patients due to reimbursement challenges [58,59].

## Interventions targeting leadership and organizational culture

This theme highlights the significant role of leadership and organizational culture in improving the health system. A few studies highlighted the significance of the interest and power of facility heads and hospital staff and political leadership in a health system. The Interventions focused on collaboration with government and non-governmental bodies and facility leaders to improve the management of diabetes. Some studies explored collaboration with facility

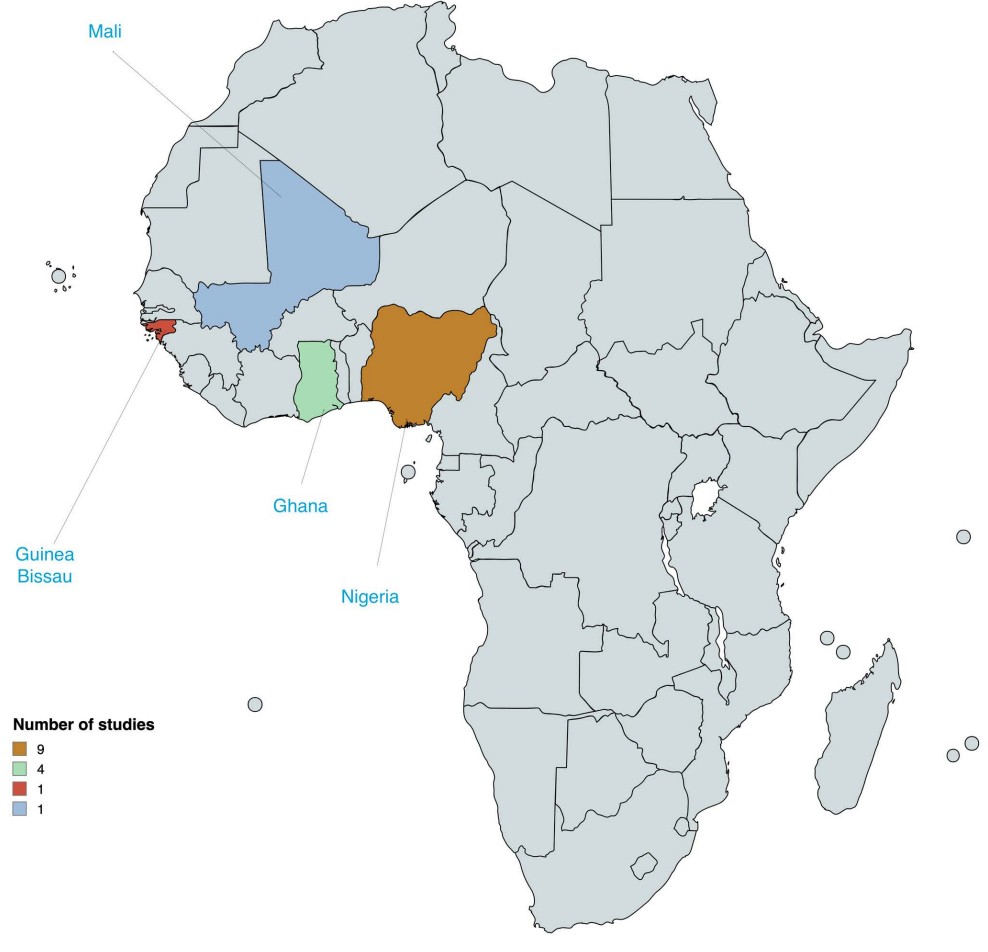

Created with mapchart.net

**Fig 4. Distribution of studies on health system interventions for type 2 diabetes in West Africa. Reprinted from [https://www.mapchart.net/] under a CC BY license, with permission from Minas Giannekas, original copyright [2025].**

leaders to provide space to set up diabetic clinics [48,60]. Another study explored collaboration between the government, non-governmental bodies and health institutions to obtain diabetic kits, provide insured medications and improve staff quality across various primary health facilities in Ghana [48]. General challenges identified include; unwillingness of hospital administrators to release space and trained personnel for the establishment of diabetes clinics; some hospital labs were unwilling to make the necessary changes in their work schedule to accommodate the newly established diabetes services; breakdown of communication between the ministry of health and the managers of the diabetes management program; several

**Table 1. Cross-cutting themes.**

| Cross-cutting theme | Sub-themes | Building blocks involved Number of studies (references) | Outcome across studies (Glycemic control, treatment adherence, disease awareness, other outcomes) | Quality of evidence on glycemic control, adherence and disease awareness using GRADE |
|---|---|---|---|---|
| Interventions targeting health worker's availability, roles and activities | • Training of service providers<br>• Forming multi-professional diabetes teams<br>• Enhancing information sharing<br>• Task sharing<br>• Increasing provider density | • Health Workforce<br>• Health Information<br>• Leadership/ Governance<br>• Supply chain<br>• 8 studies [48–51,53,54,56,60] | • Improved glycemic control<br>• Increased provider awareness to standard treatment guidelines<br>• Early detection of complications<br>• Improved patient access to skilled care. | Moderate quality (due to uncertainty of risk of bias in most studies) |
| Interventions targeting institutional infrastructure and resources for management | • Setting up diabetes clinic within health facilities<br>• Health Insurance reform<br>• Supply of equipment and materials for diagnosis and treatment | • Service delivery<br>• Supply chain<br>• Finance<br>• Leadership/ Governance<br>• 4 studies [48,58–60] | • Improved adherence to treatment plans and clinic appointments<br>• Improved glycemic control<br>• Financial risk protection | Low quality (due to indirectness and uncertainty of risk of bias in most studies) |
| Interventions targeting leadership and organizational culture | • Collaboration between government/non-governmental bodies and health facilities for lesson learning and scale up, commitment and support for type 2 diabetes control<br>• Collaboration between health facility leaders and service providers for effective operation of diabetes programs | • Leadership/ Governance<br>• Service delivery<br>• Health Information<br>• Supply chain<br>• 2 studies [48,60] | • Increased responsiveness from higher-level stakeholders<br>• Improved patient access to screening and diagnosis<br>• Higher coverage of interventions<br>• No direct measurement of glycemic control/treatment adherence | Not Applicable |
| Interventions targeting relationships among stakeholders | • Increasing provider-patient contact time<br>• Enhancing coordination/navigation of care.<br>• Strong links and communication between hospital staff<br>• Strong links and communication among patients-including peer-to-peer/group discussions | • Leadership<br>• Service delivery<br>• 6 studies [46,52,53,55,57,60] | • Improved glycemic control<br>• Increased adherence to treatment<br>• Increased knowledge and awareness of type 2 diabetes<br>• Enhanced quality and safety of type 2 diabetes services. | High quality |

*(Continued)*

**Table 1.** (Continued)

| Cross-cutting theme | Sub-themes | Building blocks involved Number of studies (references) | Outcome across studies (Glycemic control, treatment adherence, disease awareness, other outcomes) | Quality of evidence on glycemic control, adherence and disease awareness using GRADE |
|---|---|---|---|---|
| Interventions targeting empowering people (clients) | • Education on lifestyle modification<br>• Self-management education<br>• Education on chronic care | • Not highlighting building blocks<br>• 7 studies [46,47,52,53,55–57] | • Variations in impact on glycemic control (some studies showed improved glycemic control, others showed no improvement)<br>• Increased health literacy/awareness to type 2 diabetes prevention and treatment<br>• Increased adherence to treatment<br>• Reduction in stigma<br>• Improvement in health-related quality of life | Moderate quality (due to high risk of bias in most studies |

hospitals pharmacies were reluctant to purchase the diabetes kits with reason being it took a long time to sell out.

## Interventions targeting relationships among stakeholders

Although, the studies were not directly focused on building relationships among stakeholders, our analysis showed various strategies that enhanced relationships between healthcare providers and patients, healthcare providers and patient's care givers, patients and their peers, and finally health staff and health leadership. Strategies like allowing patients to receive care from the same physician over a period, increasing communication time between patients and physicians, and involving care givers during consultation were seen to be beneficial [52,55,57,60]. When patients received care from the same physician and had longer consultations, their concerns were addressed, and glycemic control was improved [57]. When care givers were involved during patient consultation, their knowledge on diabetes management was increased, stigma was bridged, patients were likely to be adherent to treatment as their families provided them with the needed support, and glycemic control was achieved compared to a control group [55]. The peer-to-peer groupings and discussions was another strategy that enhanced communication among patients. They encouraged each other to adhere to self-care practices, appointment times and sometimes discouraged colleagues from seeking traditional care [46,52,53]. The relationship between clinic leadership and their staff was improved by effective feedback systems [60].

## Interventions targeting empowering people (clients)

Seven studies explored interventions empowering people/clients to become decisive and most active participants in their diabetes care [46,47,52,53,55–57]. However, there were variations in the impact of the interventions on the outcome measures.

A pilot RCT study in Ghana, explored the effect of self-management education on glycated hemoglobin (HbA1C) levels using a one-to-one approach [47]. There was a slight decrease in HbA1C levels in both intervention and control groups after 3 months of continuous education (i.e., − 0.9% in the intervention group versus − 0.3% in the control group). This decrease was not statistically significant. In Mali however, a similar pilot RCT study which followed a grouped based approach to educate diabetic patients on self-management practices resulted in a significant improvement in HbA1C levels after 12 months [46]. The reduction in HbA1C levels between baseline and 12 months was higher (p = 0.006) in the intervention group (−1.05 [SD = 2.0]) than in the control group (−0.15 [SD = 1.7]). Additionally, patients' knowledge on the management of hypoglycemia improved after the program.

Five studies from Nigeria highlighted significant impact of highly intensive chronic care education on patients' glycemic control, treatment adherence, and quality of life [52,53,55–57].

Across all studies, the educational interventions generally consisted of first training of the health providers or making them aware of the need for health education, then one-to-one specific/individualized health education was provided to the participants to address their concerns, side effect of medications, clarification of medication doses, insulin administration, home monitoring of glucose profile, physical activity, diet, and spacing of meals. Although this theme was created during our analysis to highlight the importance of clients' power for diabetes care, interventions had some focus on improving the health workforce (i.e., through service provider training) and the service delivery building blocks of the health system.

## Discussion

This systematic review synthesizes the limited evidence on the impact of already set up interventions and pilot interventions implemented in healthcare systems to improve the availability, delivery, accessibility and quality of type 2 diabetes services as well as patient outcomes in West Africa. Within this limited body of evidence, most of the few studies available did not meet the first strict criteria for our systematic review. Nonetheless, there are ideas that can be derived from these studies about potential interventions to subject to rigorous evaluation.

Five cross-cutting themes exploring interventions targeting integrated health systems building blocks and their impact on type 2 diabetes outcomes were used to summarize the findings in this review.

The first theme highlights interventions that improved health provider's availability and their roles and activities towards type 2 diabetes care. The availability of appropriately trained health workers positively affects the health system (workforce, service delivery, leadership) and improves glycemic and non-glycemic outcomes [61,62]. The approach to increasing availability of health workers in West Africa may not be straight-forward considering constraints with deployment of health workers and reduction of attrition, willingness of available providers to take up new roles and policies restricting health providers from performing certain activities [63–65].

Studies from this review prove practical and effective ways to increase the availability of health providers and enhance their quality-of-service delivery and further correlates with improved patient outcomes. One approach is by training the available health workers in underserved areas to form a diabetes team. This was seen to be very plausible but could be undermined by challenges from the staff including unwillingness to take up new roles, demanding an increase in salaries, complaints of increased workload and lack of equipment to perform their task [48,52,53,56]. Another approach is through task sharing. The role of task sharing has been widely studied in the management of NCDs and has been noted to be an

important strategy to improve provider-patient ratio [66]. This review found non-physician health workers conducting aspects of diabetes care originally reserved for physicians, after they trained to take up such roles. The few studies that explored the impact of task sharing on patient outcomes recorded improved glycemic control and treatment compliance [49–51,54].

Institutional infrastructure and resources needed for management are essential to the robustness of a health system. The idea is not only to increase resource availability but also to make resources easily accessible for the marginalized community [67]. This theme provides insights into how multiple health system areas affect the availability, accessibility and appropriateness of physical structures, equipment and other resources needed for the management of type 2 diabetes. One approach was setting up specific days for diabetes and hypertension consultations, once a week or twice weekly. No new physical structures were built but space was given, and a day appointed solely for diabetes patients. This led to an increase in patients' adherence to clinic appointment schedules and treatment plans, mainly due to reduction in waiting times and increased physician-patient consultation time; more time was dedicated to counselling patients and addressing their concerns [48,68]. One study showed that these interventions could be altered when there was poor in-house coordination or reluctance of hospital leadership to offer space for the establishment of the diabetes clinics [48]. Interventions that focused on supply of diagnostic equipment, glucose monitoring devices and test strips, medication and other resources for management were noted to be important for the health system [48]. Health insurance, which allows patients to receive subsidized or free diabetic medications is another important intervention. Some studies on health insurance showed positive correlation with glycemic control. However, only a few patients in urban areas had access to a continuous supply of medications. Delays in reimbursement was the major backlash while inability to afford insurance packages played second hand. We found out that only two countries in West Africa supplied free/subsidized diabetes medications under their health insurance policies: Nigeria and Ghana [58,59].

The fourth theme highlights the role of leadership and organizational culture in a functioning health system towards the management of type 2 diabetes. Effective leadership is important in the implementation of health systems interventions by influencing the extent to which other inputs (e.g., trained health personnel, proper use of resources) work well in the management of chronic diseases [69]. Studies found in this review suggest that fostering continuous collaborations and clear and open line communication between facility leaders and government stakeholders can enhance the provision of resources for the management of diabetes [48,68]. Added mechanisms for evaluation and lesson-learning among stakeholders at the facility level were seen to be necessary to sustain the quality of diabetes management programs [68].

In the past and recent times, there has been evidence to support the importance of fostering relationships between stakeholders, particularly between patients and their providers, patient and their peers and hospital leadership and their staff for the management of type 2 diabetes [70–72]. Across multiple studies, effective communication and shared decision-making between providers and patients, providers and caregivers and health staff and leadership were found to be important. Effective communication between health providers and caregivers allowed caregivers to understand their client's condition and provide better support to their health care [55]. Patients who received personalized care and had their concerns addressed were more compliant with treatment compared to those who received general or usual care [57]. Peer-to-peer discussions helped reduce stigma and led to improvement on care-seeking behavior and adherence to treatment [46]. It was easy to reconfigure existing structures and routines within facilities where their leaders had fostered effective communication and feedback systems with their staff [68].

The fifth theme highlights interventions targeting patient's/client empowerment. Noticeable were education on lifestyle modification, education on self-management and education that considered the chronic dimension of diabetes. Patient/client power, preferences and interest are key determinants to a functioning health system, and one clear pathway to empower patients is through health literacy [39,68,73]. The process of promoting health literacy may sometimes become complex. People may place different emphasis on the quality of the health literacy interventions, or the tools used to implement the interventions or their longevity and reliability. High quality education for chronic diseases including diabetes follows evidence-based guidelines, uses interactive material and strategies to ensure longevity and usually leads to better health outcomes [68,73–75].

In our review, we found significant associations between education on lifestyle modification, disease complication and treatment adherence with improved glycemic control [52,55,57]. A few points are worth making note of on how these educational strategies were implemented to achieve maximum impact. First, in most of the studies with the highest impact on glycemic control, the instructors, mostly healthcare staff, were trained to deliver evidence-based guidelines [52,55,57]. Second, the education provided was culturally specific and tailored to meet individual patients' needs, including diabetes risk management, recognition of signs and symptoms of complications, medication-related education, nutrition and exercise, health care use even in the absence of symptoms, emotional and stress management, foot and skin care, and psycho-education to family members so that they could understand the effect of the illness and assist the patient [52,53,55,56]. Lastly, these empowerment-based educational sessions were interactive and persisted outside normal clinic hours; patients received both verbal and educational resource materials, they and sometimes their families were allowed to voice their concerns during sessions, provided opportunity to receive continuous education through interactive mobile appointments, record emerging issues after applying what they were taught, and receive solutions as and when needed [52,55,56].

The next empowerment-based approach was on diabetes self-management education (DSME) [46,47]. By considering the context of the illness, prevailing health practices, and the chronic dimension of the disease, this approach focused on 1) promoting patients' understanding of key concepts in their interactions with their social environment, 2) increasing their knowledge of symptoms, 3) training on glucose self-monitoring and management of hypoglycemia. The impact on glycemic control and other non-glycemic outcomes were however, varied among 2 studies found. One study which followed a grouped-based DSME intervention [46], showed improved glycemic control at the end of 12 months of implementation while another study [47] which followed a one-to-one approach, with sessions lasting for only 6 hours, showed no significant impact between the intervention group and control group after 3 months of intervention. Similar studies globally have shown varying impact on glycemic control, while mostly grouped-based interventions, showed positive impact on glycemic control. In recent systematic reviews of RCTs on the topic [76,77], it was noted that group-based DSME interventions provided a favorable effect on glycemic control compared to one-to-one approach, particularly programs conducted for 10 hours or more (high intensity programs) [78]. This is because the group-based approach allows for hands-on participation and interaction among peers to enhance individual understanding and adherence compared to a one-to-one approach [79]. Overall engagement in diabetes self-management education results in a statistically significant decrease in HbA1C levels and improves patients' self-management behavior. Our review found robust data suggesting that the high intensity, grouped-based intervention is well suited in the West African context. However, we have little understanding of how

intervention components promote behavioral modifications or lifestyle change which may help improve clinical outcomes.

## Strengths and limitations

We found pilot interventions and some already implemented interventions to enhance the capacity of type 2 diabetes management in West Africa. We proved cause-and-effect relationship between some health system interventions and patient outcomes by exploring studies with robust study designs. We provided understanding of how most of the interventions are implemented for maximum impact.

However, there is a possibility of publication bias as only published data were included in this review. The main limitation though is that most of the studies which met the inclusion criteria were under-powered and of low quality.

## Conclusion

The narratives from the literature on health system interventions for the management of type 2 diabetes in West Africa suggest patterns for strengthening or replication of health system interventions for the management of NCDs like diabetes or hypertension and can be adopted by several countries in the African region as well as Low-and-middle income countries (LMICs). The scale up of these pilot interventions may be feasible to tackle the burden of non-communicable diseases. However, there is a wide gap on comprehensive exploration of the impact of integrated system-level interventions on the patient's health outcomes and we recommend for more robust evaluation research on the impact of health system interventions at the primary care level in West Africa to inform policy and program decision making and implementation.

## Supporting information

**S1 Text. Study Protocol.**
(DOCX)

**S1 File. PRISMA checklist.**
(PDF)

**S2 File. Search Strategy.**
(PDF)

**S3 File. Quality Assessment Tools.**
(PDF)

**S1 Table. Characteristics and Quality Assessment of Studies.**
(XLSX)

**S1 Data. Table of Studies identified in Literature search.**
(XLSX)

## Acknowledgments

We would like to acknowledge the London school of Hygiene and Tropical Medicine (LSHTM) public health library for support in retrieving some studies that were not easily accessible online.

## Author contributions

**Conceptualization:** Eugene Paa Kofi Bondzie.

**Data curation:** Eugene Paa Kofi Bondzie, Kezia Amarteyfio, Yasmin Jahan, Nana Efua Enyimayew Afun, Mary Pomaa Agyekum, Ludovic Tapsoba.

**Formal analysis:** Eugene Paa Kofi Bondzie, Dina Balabanova.

**Funding acquisition:** Tolib Mirzoev, Irene Agyepong.

**Methodology:** Eugene Paa Kofi Bondzie, Kezia Amarteyfio, Yasmin Jahan, Nana Efua Enyimayew Afun, Mary Pomaa Agyekum, Ludovic Tapsoba.

**Supervision:** Dina Balabanova, Tolib Mirzoev, Irene Agyepong.

**Validation:** Dina Balabanova, Tolib Mirzoev, Irene Agyepong.

**Writing – original draft:** Eugene Paa Kofi Bondzie.

**Writing – review & editing:** Eugene Paa Kofi Bondzie, Kezia Amarteyfio, Yasmin Jahan, Nana Efua Enyimayew Afun, Irene Agyepong.

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
