## [Decision Letter · Decision Letter 0]

4 Feb 2025

Impact of health systems interventions in primary health settings on type 2 diabetes care and health outcomes among adults in West Africa: a systematic review

PONE-D-24-41758

Dear Dr. Bondzie,

We’re pleased to inform you that your manuscript has been judged scientifically suitable for publication and will be formally accepted for publication once it meets all outstanding technical requirements.

Kind regards,

André Luis C Ramalho, PhD

Academic Editor

PLOS ONE

**Additional Editor Comments** :

After a thorough review of the manuscript **"Impact of health systems interventions in primary health settings on type 2 diabetes care and health outcomes among adults in West Africa: a systematic review"** , we confirm that it fully meets the methodological and scientific standards required for publication. The first reviewer recommended direct acceptance, while the second provided minor suggestions that, while valuable, do not affect the validity, coherence, or quality of the study.

The comments from Reviewer 2 suggest methodological refinements that could enhance the manuscript but do not indicate any omissions or gaps that undermine its scientific integrity. The study adheres strictly to PRISMA guidelines and presents a well-structured strategy for assessing health system interventions in the West African context.

Therefore, no further modifications by the authors are deemed necessary before publication. The manuscript, in its current form, is already suitable for acceptance, and the points raised by Reviewer 2 may be considered in future research at the authors’ discretion. Consequently, the editorial decision is **to accept, with no required revisions** .

**Comments to the Author**

1. Is the manuscript technically sound, and do the data support the conclusions?

Reviewer #1: Yes

Reviewer #2: Yes

2. Has the statistical analysis been performed appropriately and rigorously?

Reviewer #1: N/A

Reviewer #2: N/A

3. Have the authors made all data underlying the findings in their manuscript fully available?

Reviewer #1: Yes

Reviewer #2: Yes

4. Is the manuscript presented in an intelligible fashion and written in standard English?

Reviewer #1: Yes

Reviewer #2: Yes

Reviewer #1: I believe this study is interesting and relevant in theme and scope, especially given the relevance of the topic on the impact of health systems interventions in primary health care settings on type 2 diabetes care and health outcomes among adults in West Africa. After careful consideration, I believe the manuscript is suitable for publication.

Reviewer #2: Type 2 diabetes incidence has increased, proving this to be a global public health problem that affects not only adults but also younger people. With my understanding and knowledge, the manuscript appears to be technically good. A well-structured systematic review strategy is used to investigate current published evidence on health systems interventions to support primary health facilities for type 2 diabetes care and impact on health outcomes, service access and quality in West Africa. The methodology follows existing systematic review procedures, namely the PRISMA checklist, which is widely accepted in this field of knowledge. However, some insights might need to be addressed, namely:

- Regarding Search Methods section, authors mention “Using the following keywords and their medical subject headings (…) and intersected with type 2 diabetes”. It would be interesting to understand whether authors did calibrate the query through a sensitivity analysis of the used terms instead of simply identifying and joining keywords and index terms in a search expression.

- Besides, in Study selection section, authors transported literature search results to Rayyan software “and the screening of titles and abstracts of articles were conducted by a pair of independent reviewers”, with conflicts being “resolved through discussions”. It would be interesting to know the extent of agreement between investigators in these phase (for example, by calculating the percentage calculation of agreement or Cohen’s kappa coefficient (which is a robust statistical measure of agreement)).

- Also, it would have been interesting if the authors had carried out a brief search for non-indexed databases, including in the article brief notions about the importance/uselessness of exploring this type of resources in the analysis of the topic in question, even though this was not the scope of this research.

**Do you want your identity to be public for this peer review?** For information about this choice, including consent withdrawal, please see our Privacy Policy

Reviewer #1: **Yes: ** Abel Silva de Meneses

Reviewer #2: **Yes: ** Bruno Filipe Coelho da Costa

---

## [Editor Report · Acceptance letter]

PONE-D-24-41758

PLOS ONE

Dear Dr. Bondzie,

I'm pleased to inform you that your manuscript has been deemed suitable for publication in PLOS ONE. Congratulations! Your manuscript is now being handed over to our production team.

Kind regards,

on behalf of

Prof. Dr. André Luis C Ramalho

Academic Editor

PLOS ONE